# Differences of Rainfall–Malaria Associations in Lowland and Highland in Western Kenya

**DOI:** 10.3390/ijerph16193693

**Published:** 2019-09-30

**Authors:** Naohiko Matsushita, Yoonhee Kim, Chris Fook Sheng Ng, Masao Moriyama, Tamotsu Igarashi, Kazuhide Yamamoto, Wellington Otieno, Noboru Minakawa, Masahiro Hashizume

**Affiliations:** 1Department of Paediatric Infectious Diseases, Institute of Tropical Medicine (NEKKEN), Nagasaki University. Nagasaki 852-8523, Japan; hashizum@nagasaki-u.ac.jp; 2Graduate School of Biomedical Sciences, Nagasaki University, Nagasaki 852-8523, Japan; 3Department of Global Environmental Health, Graduate School of Medicine, The University of Tokyo, Tokyo 113-0033, Japan; yoonheekim@m.u-tokyo.ac.jp; 4School of Tropical Medicine and Global Health (TMGH), Nagasaki University, Nagasaki 852-8523, Japan; chrisng@nagasaki-u.ac.jp; 5Division of Electrical Engineering and Computer Science, Graduate School of Engineering, Nagasaki University, Nagasaki 852-8521, Japan; matsu@cis.nagasaki-u.ac.jp; 6Remote Sensing Technology Center of Japan (RESTEC), Tokyo 105-0001, Japan; igarashi_tamotsu@restec.or.jp; 7Japan Aerospace Exploration Agency (JAXA), Tokyo 101-8008, Japan; yamamoto.kazuhide@jaxa.jp; 8Centre for Research and Technology Development Maseno University, Kisumu 40100, Kenya; restechmaseno@yahoo.com; 9Department of Vector Ecology and Environment, Institute of Tropical Medicine, Nagasaki University, Nagasaki 852-8523, Japan; minakawa@nagasaki-u.ac.jp

**Keywords:** time-series analysis, distributed lag nonlinear model (DLNM), lagged effect, heterogeneity

## Abstract

Many studies have reported a relationship between climate factors and malaria. However, results were inconsistent across the areas. We examined associations between climate factors and malaria in two geographically different areas: lowland (lakeside area) and highland in Western Kenya. Associations between climate factors (rainfall, land surface temperature (LST), and lake water level (LWL)) and monthly malaria cases from 2000 to 2013 in six hospitals (two in lowland and four in highland) were analyzed using time-series regression analysis with a distributed lag nonlinear model (DLNM) and multivariate meta-analysis. We found positive rainfall–malaria overall associations in lowland with a peak at 120 mm of monthly rainfall with a relative risk (RR) of 7.32 (95% CI: 2.74, 19.56) (reference 0 mm), whereas similar associations were not found in highland. Positive associations were observed at lags of 2 to 4 months at rainfall around 100–200 mm in both lowland and highland. The RRs at 150 mm rainfall were 1.42 (95% CI: 1.18, 1.71) in lowland and 1.20 (95% CI: 1.07, 1.33) in highland (at a lag of 3 months). LST and LWL did not show significant association with malaria. The results suggest that geographical characteristics can influence climate–malaria relationships.

## 1. Introduction

Malaria is a life-threatening febrile disease caused by Plasmodium parasites transmitted by vector mosquitoes [1]. In 2017, about 219 million cases occurred globally and 435,000 died of malaria, and nearly half of the world’s population is at risk of malaria [1]. A total of 92% of malaria cases and 93% of malaria deaths occurred in the WHO Africa region [1]. 

Malaria has been endemic in Western Kenya but spatially heterogeneous between lowland and highland areas [2,3]. The malaria incidence has been consistently high in lowland areas [2], whereas there has been long-term fluctuation of malaria incidence in highland areas. Until the 19th century, there was no significant health burden by malaria in highland [2]. However, the first malaria epidemic occurred after the First World War in 1918–1919 [2]. After that, the Western Kenyan highlands experienced malaria epidemics infrequently from the 1920s to the 1950s and malaria-free periods from the 1960s to the early 1980s. However, malaria reemerged in this region in the 1980s, and epidemics have been often reported since that time [2,4,5].

Geographical differences between the lowland and highland areas could play a major role in the spatial heterogeneity in malaria incidence. The lowland area in Western Kenya is adjacent to Lake Victoria. The climate in this area is relatively hot and dry and tends to develop many swamps during a rainy season [6]. By contrast, the highland areas with an altitude of more than 1500 m above sea level are located on the eastern side of the lowland area with moderate climates (daily mean temperatures of below 20 °C) that support farmland and forest [2,4].

Many studies have been conducted to evaluate relationships between climate factors and malaria transmission in Africa and other regions [3,7,8,9,10,11,12,13,14,15,16,17,18,19,20,21,22]. The reported associations were not consistent across the areas, which may suggest that geographical characteristics or other local factors influenced climate–malaria relationships [10,23,24]. For example, a study in China showed positive and negative rainfall–malaria associations depending on locations, although the causes of the heterogeneity were not discussed [10]. In a study in Ethiopia, the lag pattern of rainfall–malaria association exhibited slight heterogeneity. Rainfall had positive association in both hot districts with a lower altitude and cold districts with a higher altitude. However, the effect of rainfall declined after considering longer lags in the hot district, which could be explained by the drying up of breeding sites due to higher temperatures [23].

Recent studies conducted in lowland area in Western Kenya have reported associations among temperature, rainfall, and malaria [3,13]. Sewe et al. reported positive associations between rainfall and malaria in locations near Lake Victoria. Several studies have also examined relationships of climate factors such as rainfall, temperature, and global climate variabilities (e.g., Indian Ocean Dipole) with malaria in the highlands [25,26,27,28]. However, to our best knowledge, no study has systematically quantified the role of climate factors on malaria incidence across multiple locations characterized by different geographical features including both lowland and highland areas.

In this study, we aimed to examine associations between climate factors and malaria in two different malaria endemic areas, the lowlands and highlands of Western Kenya, using a distributed lag nonlinear model (DLNM) to examine delayed effects and nonlinear relationships [29]. We attempted to clarify how the influence of climate factors differs between mountainous and forested areas, like the highlands, and swampy areas, like the lowland areas along Lake Victoria. These relationships might provide insights into how geographical characteristics or local factors influence climate–malaria relationships.

## 2. Methods

### 2.1. Study Area

The study area of lowland and highland in Western Kenya is in the Nyanza and Rift Valley provinces with an area of about 195,000 km^2^ and population of about 15 million in total [30]. Malaria is endemic to both areas, although those two areas are geographically contrasting (Figure 1B).

### 2.2. Case Data

The number of monthly malaria cases was obtained from hospitalization and consultation ledgers of 6 hospitals. The hospitals were classified into two areas by altitude: The lowland below 1500 m were Nyanza and Kendu Bay, and the highland above 1500 m were Maseno, Kisii, Kericho, and Kapsabet (Table 1). Cases were hospitalized cases in Nyanza, Kendu Bay, and Maseno; outpatient cases in Kisii, Kericho, and Kapsabet. Different types of data—hospitalized and outpatient cases—were collected in this study because each hospital had only one type of data. The period of the case data collected for this study was from 10 years to 14 years in 2000–2013 depending on the hospitals.

### 2.3. Environmental Data

Remotely-sensed daily rainfall data were obtained as Global Satellite Mapping of Precipitation (GSMaP) data from the Remote Sensing Technology Center of Japan (RESTEC) and Japan Aerospace Exploration Agency (JAXA) [31]. Spatial resolution of the data was a lattice with grid latitudes and longitudes of 0.1 degrees, representing approximately 10 km square around the study area. The rainfall data included the location of each hospital providing case data. The data were integrated to provide monthly cumulative rainfall figures.

Remotely-sensed daily land surface temperature (LST) and normalized difference vegetation index (NDVI) data were obtained from the moderate resolution imaging spectroradiometer (MODIS) sensors aboard the National Aeronautics and Space Administration (NASA) satellites [32]. Spatial resolutions of the data were 1 km square for LST and 250 m square for NDVI. Data were integrated to 8 km radius data around each hospital calculating spatial means in the radius. Daily LST and NDVI data were converted to monthly mean data.

Lake water level (LWL) data were obtained from the United States Department of Agriculture (USDA) [33]. The raw data were Lake Victoria height variations every 10 days, which were converted to monthly mean data.

### 2.4. Statistical Analysis

We modeled lagged relationships among rainfall, LST, and monthly malaria cases in each hospital using a distributed lag nonlinear model in a quasi-Poisson regression framework, controlling for long-term trend and seasonally varying factors [29]. We allowed lags of up to 6 months (lags 0–6) for rainfall and 3 months (lags 0–3) for LST, which were enough to capture attenuation of the effects based on initial analysis incorporating longer lags and biological causality between those exposures and malaria, considering time span from the appearance of vector mosquito breeding sites due to rainfall or influences on the mosquito life cycle by temperature and rainfall to increase or decrease in malaria cases.

To account for seasonality and long-term trend, we incorporated two natural cubic splines: one for the month of year and another for year with degrees of freedom (*df*) guided by quasi-Akaike’s information criterion (QAIC). *df*s were selected by this approach for each location.

The model equation was:
Log[E(Y*_t_*)] = α+ *f*(R*_t_*) + *f*(LST*_t_*) + ns(*moy*,df) + ns(*year*, df) [1](1)
where Y*_t_* represents the monthly number of malaria case in month *t*, αis the intercept, *f*(R*_t_*) and *f*(LST*_t_*) are the matrices obtained by applying a cross-basis function to rainfall and LST based on DLNM, respectively. “ns” is a natural cubic spline; *moy* is month of year; *df* denotes degree of freedom; *year* represents integer 2000 to 2013.

Considering possible involvement of other environmental factors in relationships among rainfall, LST, and malaria, we conducted additional subgroup analyses using a model incorporating LWL. We allowed lags of up to 6 months (lags 0–6) for LWL based on similar approach to that of the other exposures. The model incorporating LWL was applied only in the lowland area because the water level of Lake Victoria was unlikely to affect malaria in the highlands. NDVI was omitted from the model because NDVI had little effects on malaria and little role in any relationships among rainfall, LST, and malaria in the initial analyses.

The model equation was:
Log[E(Y*_t_*)] = α+ *f*(R*_t_*) + *f*(LWL*_t_*) + *f*(LST*_t_*) + ns(*moy*,df) + ns(*year*, df) [2](2)
where *f*(LWL*_t_*) is the matrices obtained by applying a cross-basis function to lake water level through DLNM, respectively.

We estimated relative risks (RRs) of malaria for rainfall, LST, and LWL at each lag with reference to 0 mm, 22 °C and −1.0 m, respectively. Basically, minimum values of exposures were chosen as reference values. As for LST, the minimum value in Nyanza, the hottest location was chosen as the reference value because of a large variation of LST among hospitals. We plotted the RRs of malaria for rainfall, LST, and LWL over lags of 0 to 6, 0 to 3 and 0 to 6 months, respectively. Then, we performed meta-analysis by area: the lowland (Nyanza and Kendu Bay) and the highland (Maseno, Kisii, Kericho and Kapsabet) in the model [1] (the main model) [34]. We pooled the hospital-specific rainfall–malaria associations by area. We plotted RRs of malaria for rainfall over lags of 0 to 6 months. We also estimated pooled lag-specific and predictor-specific rainfall–malaria associations by area. We plotted RRs at lags of 0, 3, and 6 months for lag-specific associations, and at 50, 150, and 300 mm for rainfall-specific associations.

As for LST–malaria associations, we estimated pooled associations of all the hospitals irrespective of elevation due to lack of a difference between lowland and highland. We plotted RRs of malaria for LST over lags up to 3 months.

For the model [2] (the submodel), we estimated pooled overall rainfall–malaria, LST–malaria, and LWL–malaria associations in lowland area only. We plotted RRs of malaria for rainfall, LST, and LWL over lags of up to 6, 3 and 6 months, respectively.

We assessed the sensitivity of our results by changing the *df* (2 to 4) for the month-of-year spline and lags of rainfall (4 to 8) included in the model.

Analysis was conducted using R version 3.6.0 (R foundation for Statistical Computing, Vienna, Austria).

## 3. Results

Time-series plots of malaria cases and exposures in each location are shown in Figure 2 and Appendix A. In the lowland area, temporal declines in malaria cases were observed from 2006 to 2008 in Nyanza, in 2007 in Maseno, and from 2003 to 2007 in Kendu Bay (Figure 2 and Appendix A). In Nyanza, Kendu Bay, and Maseno, the number of cases was much smaller than in the others, because they were data of hospitalized patients.

In general, high temperature and high amount of rainfall were observed in Nyanza, Kendu Bay, and Maseno, whereas mild temperature and lower amount of rainfall were observed in Kericho and Kapsabet. Mild temperature and high amount of rainfall were observed in Kisii (Figure 2 and Appendix A and Table 1).

The pooled overall rainfall–malaria associations by area (lowland and highland areas) are shown in Figure 3. Positive associations were observed over all the rainfall level in lowland. RR peaked at approximately 120 mm per month of rainfall, and the RR was 7.32 (95% CI 2.74, 19.56) (Appendix A). RR decreased at higher rainfall level. No evidence of association with rainfall was observed in highland.

The pooled lag-specific rainfall–malaria associations by area are shown in Figure 4. Positive associations were observed almost up to 300 mm of rainfall at lags 2 to 4 in both lowland and highland, although a wider range of the lagged effects was observed from 1 to 5 in lowland. Shapes of RR curves for each lag were similar to the RR curve of pooled overall rainfall–malaria associations. In general, RR peaked at lag 3 and rainfall of 150 mm. The RR was 1.42 (95% CI 1.18, 1.71) in lowland (Appendix A). RRs did not decrease at higher rainfall level in highland opposed to lowland (Figure 4). The RR at lag 3 and rainfall of 150 mm was 1.20 (95% CI 1.07, 1.33) in highland (Appendix A), whereas negative associations were observed at lag 6 in highland.

The pooled predictor-specific rainfall–malaria associations by area are shown in Figure 5. In lowlands, the significant positive associations were observed over lags 1 to 5 around 150 mm of rainfall. In highland, positive associations were observed at lags 2 to 4 around 150 mm of rainfall.

The estimated association between rainfall and malaria, and the lagged pattern at each location are shown in Appendix A. 

LST–malaria associations were not significant both in pooled overall analysis and pooled predictor-specific analysis (Appendix A). The RR tended to decrease at longer lags at higher LST, although not significantly (Appendix A).

The associations among rainfall, LST, LWL, and malaria by applying the LWL adjusted model (Model 2) in the lowland area only are shown in Appendix A.

After being adjusted for LWL, lag patterns of pooled rainfall–malaria and LST–malaria relationships did not change so much.

Pooled LWL–malaria associations were not significant. 

In sensitivity analyses for rainfall–malaria associations, pooled overall, lag-specific, and predictor-specific associations by area did not change substantially when changing *df* for month of year between 2 and 4 and changing *df* for rainfall lag in the model between 4 and 8 (Appendix A). 

## 4. Discussion

In this study, we investigated nonlinear and lagged relationships between climate factors (rainfall, LST, and lake water level) and malaria in Western Kenya, including locations in the lowland area around Lake Victoria and the East African highlands.

We found positive associations between rainfall and malaria in lowland, while we did not find similar associations in highland.

In lowland, pooled overall rainfall–malaria associations were positive over all the rainfall level. The relative risk (RR) increased as monthly rainfall increased up to 120 mm and gradually decreased thereafter. The positive associations and shapes of RR curves were preserved in lag-specific associations at lags 1 to 5. As for the pooled predictor-specific rainfall–malaria associations, positive associations were observed over lags 0 to 5. RR curve was a gentle inverse U shape which peaked at lag 3. The positive association could be explained by the roles of rainfall that creates swamps which can become vector mosquito breeding sites [35]. Lags might reflect time span from increase in rainfall to increase in malaria cases through mosquito larva development, biting activities, and incubation period in human body. Attenuation of the positive association at higher rainfall level might be explained by flush-out of breeding sites by flooding because many swamps are thought to exist along the lake coast [35,36]. Similar findings have been shown in a previous study near this location [3,7] and in China [9].

By contrast, in highland, pooled overall rainfall–malaria associations were not significant over all the rainfall level. It might suggest that breeding sites in areas apart from the lake coast would be less influenced by rainfall. However, in the pooled lag-specific and predictor-specific rainfall–malaria associations, positive associations were observed over lags 2 to 4, although RRs were smaller than those in lowland. The RR curve of predictor-specific associations was an inverse U shape which peaked at lag 3 like that of lowland. Similar findings were reported in previous studies in the highlands. Hashizume et al. reported a positive association between rainfall and malaria at a lag of 2–3 months in Western Kenya highlands [26]. Chaves et al. showed that positive seasonal association between rainfall and malaria decreased with altitude [28]. It might be possible that there were vector mosquito breeding sites which can easily be generated and expanded by rainfall also in highland, for example, breeding sites around irrigation canals beside farmlands [37]. Attenuations of positive associations were not observed in lag-specific associations, which may support that the flush-out effect on breeding sites by heavy rainfall was weak in the highland. Negative associations were observed at a lag of 6 months in highland. It might be explained by a temporal reduction of susceptible population due to increase in cases in previous months. However, a biological or ecological mechanism that leads to the results is not clear, which would require further studies.

As for the LST–malaria relationship, a significant association was not observed. In previous studies, relationships between temperature and malaria variated. The LST–malaria relationship varied between the areas also in a study near this location [3,38], whereas a positive association between ambient temperature and malaria was reported in several studies in China [10,11,39]. It is not clear why a significant LST–malaria association was not found in this study, although temperature might influence the larval development and biting activities of mosquitoes, as previous studies reported. It could be an issue requiring further studies.

After adjusting for lake water level, a positive association between rainfall and malaria remained, whereas negative associations were observed between LST and malaria at longer lags over all LST levels. Lake water level had no significant association with malaria.

The current study has several limitations. First, malaria cases in the study were diagnosed clinically. In malaria endemic areas like this study sites, febrile illnesses are likely to be clinically diagnosed as malaria. Thus, cases in this study might include overdiagnosed cases. However, some of the cases in the hospitals in highland, Kisii, Kericho, and Kapsabet, were also laboratory-confirmed by microscopy in part of the study period. Laboratory-confirmed cases and clinically-diagnosed cases had similar patterns of monthly variations, which might support the reliability of the clinically-diagnosed cases in the study for evaluating rainfall–malaria associations. Second, characteristics of disease data were different among the hospitals. While hospitalized cases were used in Nyanza, Kendubay, and Maseno, outpatient data were used in Kisii, Kericho, and Kapsabet. This might influence the results a little, for example, via difference of time span from onset to hospital admission and hospital visit, and via difference of severity between hospitalized and outpatient cases. However, it is not likely to affect relationships between climate factors and malaria much in the study. Third, the unit of the data in this study was monthly, which might not be able to capture more finely delayed relationships. Finally, the current study did not consider influences of interventions. Interventions such as distribution of insecticide bed net and indoor spraying can influence malaria trends [40,41,42,43]. However, influences of any intervention could be partly adjusted because we incorporated the term for controlling the long-term trend in the model.

## 5. Conclusions

This study found that, in lowland along the lake coast, rainfall and malaria had a positive association which became strong as rainfall increased up to a certain amount of rainfall, then attenuated at a higher amount of rainfall. By contrast, in highland, a similar association was not observed.

Our results suggest that geographical characteristics can influence or possibly even determine the structure of climate–malaria relationships, which might provide some ideas for understanding relationships between environmental factors and the disease.

## Figures and Tables

**Figure 1 ijerph-16-03693-f001:**
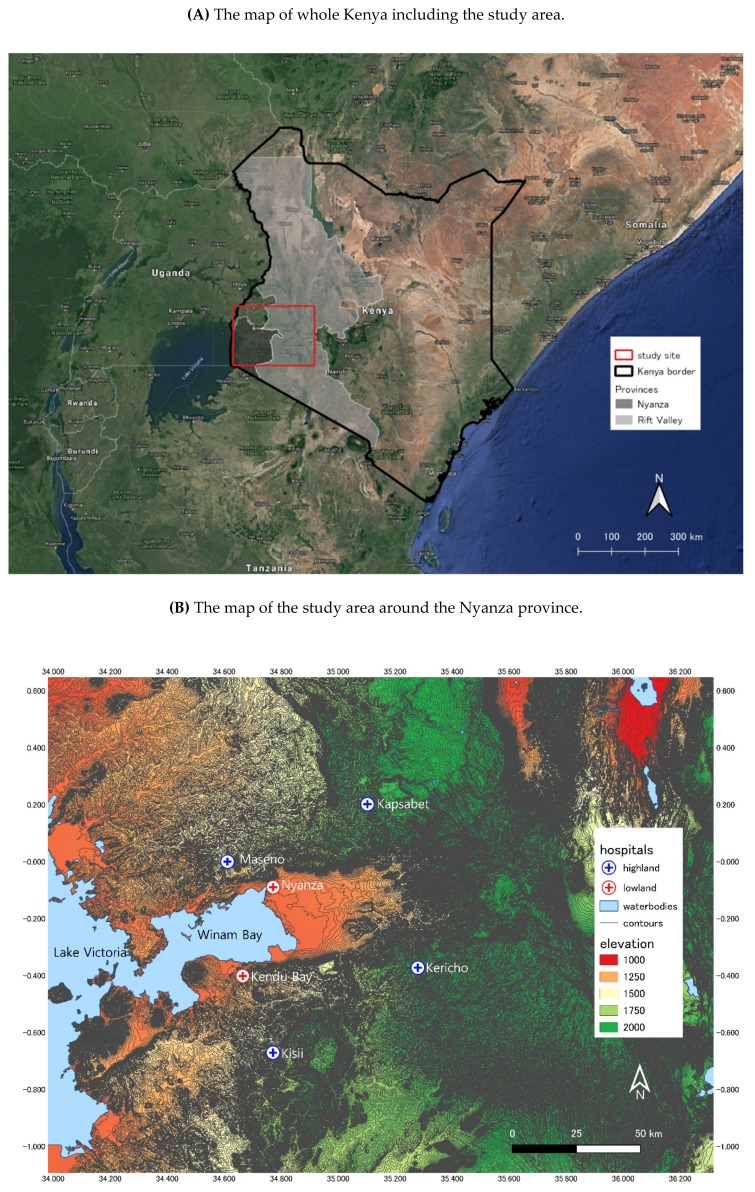
The map of the study area.

**Figure 2 ijerph-16-03693-f002:**
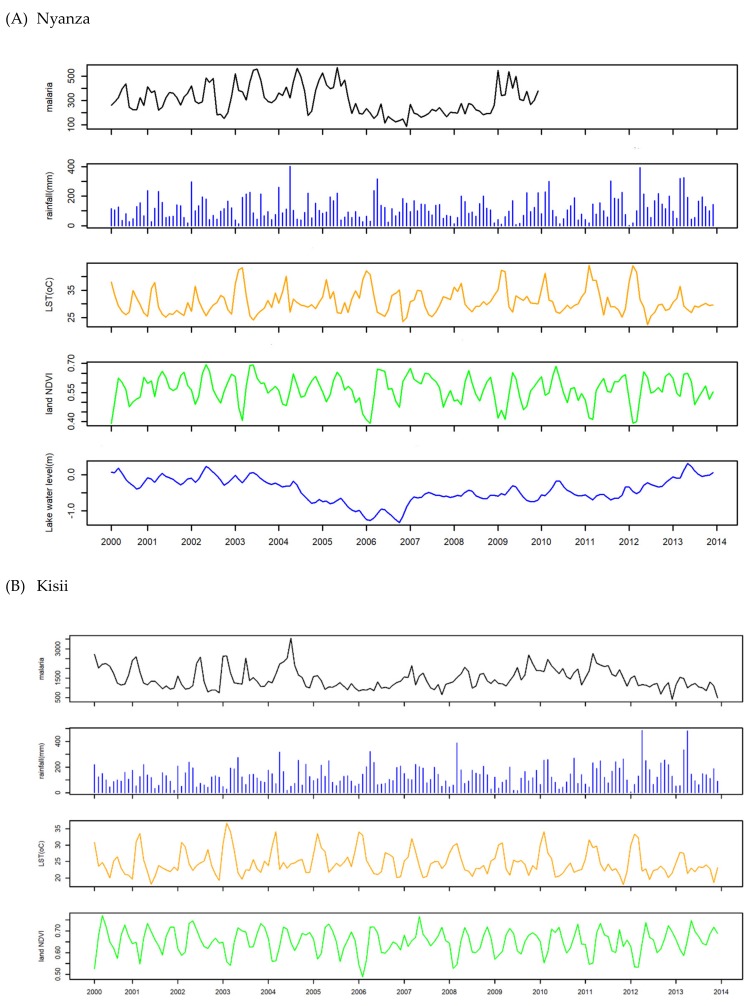
Monthly time-series plots of malaria cases, rainfall, LST, NDVI, and LWL in lowland (Nyanza) and highland (Kisii).

**Figure 3 ijerph-16-03693-f003:**
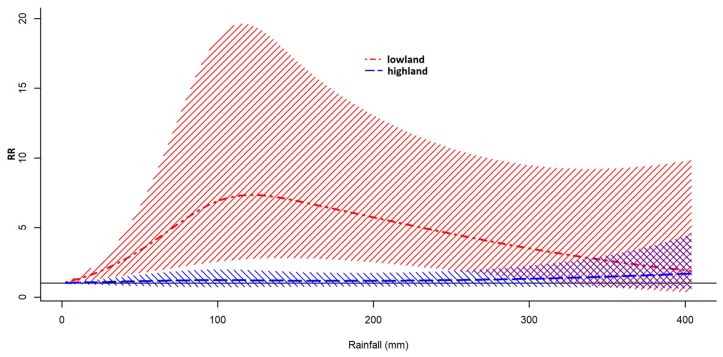
Pooled overall rainfall–malaria associations by area.

**Figure 4 ijerph-16-03693-f004:**
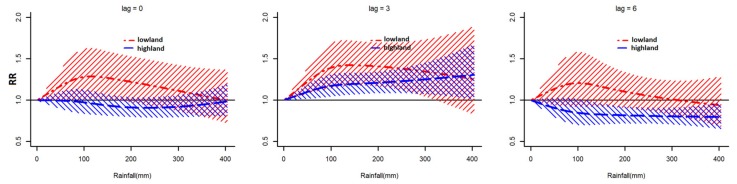
Pooled lag-specific rainfall–malaria associations by area.

**Figure 5 ijerph-16-03693-f005:**
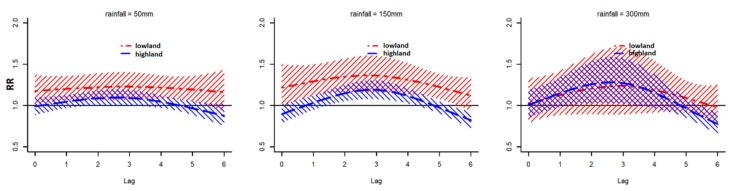
Pooled predictor-specific rainfall–malaria associations by area.

**Table ijerph-16-03693-t001a:** (**A**)

Hospital	Elevation	Period	Total	Mean	Sd	Min	Max
Nyanza	1189 m	2000–2009	36098	305.92	118.03	84	574
Kendu Bay	1243 m	2000–2009	6007	50.91	30.68	2	124
Maseno	1576 m	2000–2009	3880	32.88	17.80	2	81
Kisii	1656 m	2000–2013	244277	1471.55	528.54	407	3554
Kericho	1983 m	2001–2013	166761	1068.98	498.08	420	4969
Kapsabet	1997 m	2001–2010	157866	1315.55	691.20	387	3231

Hospitalized cases in Nyanza, Kendu Bay and Maseno: outpatients in Kisii, Kericho and Kapsabet.

**Table ijerph-16-03693-t001b:** (**B**)

Hospital	Mean	Sd	Min	Max
Nyanza	119.21	78.77	0.60	403.28
Kendu Bay	127.01	81.90	1.29	445.25
Maseno	146.92	87.19	3.15	404.44
Kisii	136.25	82.40	7.05	485.60
Kericho	89.11	61.73	0.76	371.03
Kapsabet	82.31	61.83	1.13	315.25

**Table ijerph-16-03693-t001c:** (**C**)

Hospital	Mean	Sd	Min	Max
Nyanza	30.95	4.55	22.46	44.07
Kendu Bay	31.46	5.20	20.70	47.08
Maseno	28.21	4.41	20.25	41.75
Kisii	24.66	3.81	18.02	36.74
Kericho	23.37	4.08	9.43	37.01
Kapsabet	26.14	3.56	12.23	36.36

**Table ijerph-16-03693-t001d:** (**D**)

Hospital	Mean	Sd	Min	Max
Nyanza	0.57	0.07	0.39	0.70
Kendu Bay	0.55	0.08	0.35	0.67
Maseno	0.61	0.06	0.39	0.72
Kisii	0.65	0.06	0.49	0.77
Kericho	0.70	0.04	0.57	0.79
Kapsabet	0.67	0.06	0.47	0.77

**Table ijerph-16-03693-t001e:** (**E**)

Mean	Sd	Min	Max
−0.42	0.35	−1.33	0.33

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
