# Peer review of "Differences of Rainfall–Malaria Associations in Lowland and Highland in Western Kenya"

_ijerph, 2019, doi:10.3390/ijerph16193693_

Round 1

Reviewer 1 Report

Overall the study was well designed and provides new information on the relationship between malaria and rainfall amounts. While thorough in explanation of the methodologies, some of the introduction information was vague and lacked enough detail. Similarly, the discussion focused on rainfall, while two other parameters were measured, LST and LWL. While LWL had previously little effect on malaria associations, LST has been identified as a key factor in malaria transmission. Additionally, given global climate changes the authors may want to address the possibility of potential changes in transmission given how rainfall and temperature patterns may change the transmission in both lowland and highland areas.

Specific Comments:

Line 41: Vague on where the 219 million cases occurred, please be more specific (even if you do mean globally)

Lines 61-64: Again the descriptions of previous studies are vague and leave the reader wondering why these studies were used as examples. What was it about the study in China and the different locations? What about the study in Ethiopia, was the different in temperature due to altitude? These would be highly relevant to your study and provide reader with more context.

Lines 147-148: In choosing your reference values, you mention they were determined by visual assessment. What does that mean? Reference values are important and this description is vague in its determination.

Line 149: Do you mean RRs were determined for rainfall at a lag of 6 months, LST at a lag of 3 months, and LWL at a lag of 6 months? The way it is written now is somewhat confusing. Please provide clarity.

Lines 249-252: The discussion of LST-malaria relationship is lacking. Yes, there was no significance in your study, but there has been numerous studies on the relationship between temperature and malaria transmission. I would expand this discussion and perhaps postulate as to why there was no significance among you study results.

Lines 262-267: In your discussion of hospitalization versus hospital outpatient. You do not discuss severity of disease – would this be a factor in a hospitalization versus an outpatient, were some cases in the lowlands not captured due to the severity not being enough to be hospitalized. You mention that the distinction between hospitalization and outpatient hospital visit will not affect the relationship, though there are likely different factors in a hospitalization versus an outpatient visit and this needs to be expanded upon.

Reviewer 2 Report

This research is a very good and quite difficult work as well. I also could
feel the challenge of the authors struggling to find a best explanation of
the data and graphics. Not easy at all.

There is important to note the high complex relation between rainfall-
temperature-vector evolution-local topography aspects. I need to agree
with the authors about the heavy influence of topographical aspects.
These topographical aspects make influence between the malaria case
aspects at lowland and at highland areas of Western Kenya. A good shot
to deeper this complex explanation between lowlands, highlands and
malaria cases in this study area is maintaining the focus on the optimum
environment that the vector could have, and this optimum environment
has very different aspects in lowland and in highland areas (it is two
different mosquito universes with the same elements of analysis). What is
the optimum rain volume to the vectors population between these two
areas? What is the optimum LSTs to the vectors population between these
two areas? What is the best time length for the optimum vector weather
between lowlands and highlands areas? That is good to remember that
mountainous landscapes are extremely heterogeneous which reflects the
high difficult level to develop a rational comparison between the historical
malaria events at Kenya (this situation likely could show an unique,
endemic environment to that disease if comparing with another studies at
another places of the world).

Another two observations:

MAP (FIGURE 1): Although it is an interesting map, the authors should
improved it: 1) There is not an identification of Nyanza region through
Kenya Territory; 2) I really rather the traditional geographical coordinate
system at the map (latitude and longitude); 3) please insert the north
direction at the map; 4) why the reason of this kind of colour graduation
at this map? 5) could the authors insert the geographical coordinate in
each city or village at this map (lat., long., alt.)?

Lines 90-91:"Different types of data were collected in this study because
each hospital had only one type of data". - What types of data exist?
